# Sessile Lifestyle Offers Protection against Copper Stress in *Saccharolobus solfataricus*

**DOI:** 10.3390/microorganisms11061421

**Published:** 2023-05-27

**Authors:** Alejandra Recalde, Gabriela González-Madrid, José Acevedo-López, Carlos A. Jerez

**Affiliations:** 1Laboratory of Molecular Microbiology and Biotechnology, Department of Biology, Faculty of Sciences, University of Chile, 8330111 Santiago, Chile; alejandra.recalde@biologie.uni-freiburg.de (A.R.); gabriela.gonzalez.m@ug.uchile.cl (G.G.-M.); jacevedol.1994@gmail.com (J.A.-L.); 2Molecular Biology of Archaea, Institute of Biology II-Microbiology, University of Freiburg, 79104 Freiburg, Germany

**Keywords:** biofilm, copper, Sulfolobales, metal resistance, stress response, bioleaching

## Abstract

Some archaea from the genus Sulfolobus are important for bioleaching of copper, where metal resistant microorganisms are required. Biofilm generation is one of the ways microorganisms cope with some stimuli in nature, including heavy metals. The response to external factors, particularly in the biofilm form of life, is still underexplored in archaea. To explore how model thermoacidophilic archaeon *Saccharolobus solfataricus* faces copper stress during this lifestyle, changes in biofilms were studied using crystal violet staining, confocal fluorescence microscopy, and qPCR approaches. It was found that biofilm formation reached a maximum at 0.5 mM Cu, before starting to decrease at higher metal concentrations. The morphology of biofilms at 0.5 mM Cu was observed to be different, displaying lower thickness, different sugar patterns, and higher amounts of cells compared to standard growing conditions. Furthermore, *copA*, which is responsive to intracellular Cu concentration, was downregulated in biofilm cells when compared with planktonic cells exposed to the same metal concentration. The latest results suggests that cells in biofilms are less exposed to Cu than those in planktonic culture. In a PolyP-deficient strain, Cu was not able to induce biofilm formation at 0.5 mM. In summary, the findings reported here suggest that the biofilm form of life confers *S. solfataricus* advantages to face stress caused by Cu.Biofilm formation remains a relatively unexplored topic in archaeal research. Therefore, this knowledge in model organisms such as *S. solfataricus*, and how they use it to face stress, could be of great importance to engineer organisms with improved capabilities to be applied in biotechnological processes, such as bioleaching of metals.

## 1. Introduction

Biofilms are a predominant form of life for microorganisms in nature. These communities are formed by cells attached to a surface, which can work or not as a feeding substrate [1,2]. Sessile cells are surrounded by a matrix made of extrapolymeric substances (EPS) secreted by themselves. The biofilm matrix is composed of lipids, polysaccharides, extracellular nucleic acids, and proteins [3]. Cells present in biofilms are dynamic, as they can detach and disperse in the liquid phase to live in a planktonic form, or attach again to other surfaces to colonize new sites.

The first biofilm described in archaea was that of *Thermococcus litoralis* [4]. In recent years, biofilms from Sulfolobales species have been described [5]. It was seen that changes in pH and temperature can induce biofilm formation in these Crenarchaeota, and a method to obtain mature biofilms, using plastic Petri dishes as a surface, was described [5,6].

Three steps have been described in biofilm formation: adherence to a surface in an irreversible manner, microcolony formation, and maturation and dispersion [7]. Regulation of these phenomena have been studied less in archaea than in bacteria. Bacteria secrete molecules called autoinductors (AIs) that trigger biofilm formation through regulation of changes in gene expression [1]. While homologs to proteins involved in the synthetic pathway for these compounds have been found in methanoarchaea and mesophilic haloarchaea, this is not the case for thermophilic or hyperthermophilic archaea [1]. In Sulfolobus spp., no AI molecules or similar ones have been found [3]. Widely distributed secondary messenger c-di-GMP is another molecule involved in biofilm formation in bacteria, but so far there is no evidence of its presence in archaea [7,8].

A family of Lrs-14 transcriptional regulators is known to participate in the regulation of biofilm formation in Sulfolobales [6,9]. In the archaeal thermoacidophile *Sulfolobus acidocaldarius*, six homologs to Lrs-14 were found, three of which influence biofilm formation, as well as motility [9]. One of these homologs, a protein known as archaeal biofilm regulator 1, or AbfR1 (*saci_0446*), acts as a repressor for biofilm formation, since its deletion favors EPS production [9]. Additionally, this Δ*abfR1* deletion mutant showed diminished transcriptional levels of genes coding for the archaellum, and cells were nonmotile [9]. On the other hand, elevated transcriptional levels of the *aap* (archaeal adhesive pili) operon, important for adherence to surfaces, were seen [9]. AbfR1 is regulated by phosphorylation, and is also able to regulate its own expression by binding to its promoter [10].

In addition to transcriptional regulators, there are other levels of regulation when changing lifestyle, such as noncoding sRNA and glycosylation of both pili and archaella [7].

Lewis et al. reported that type II antitoxin VapB14 regulates biofilm dispersal in *S. acidocaldarius* through traditional toxin neutralization, but also by employing noncanonical transcriptional regulation [11]. *S. acidocaldarius vapB14* and its homolog in thermoacidophile *Metallosphaera sedula* (*msed_0871*) were both upregulated in biofilm cells, supporting the role of an antitoxin in biofilm regulation [11].

Recently, it was found that the absence of PolyP impairs biofilm formation in *S. acidocaldarius* and *S. solfataricus* by affecting archaellum expression and assembly, as well as cell motility and adhesion [12].

Biofilms confer advantages to cells, as they are involved in resistance to different types of environmental stress, such as desiccation, pH changes, UV radiation, and high salt or metal concentrations [2,13,14]. In the context of metal resistance, biofilms can passively trap metals by immobilizing or even precipitating them. This occurs primarily because most components within the extracellular matrix possess a negative charge [14]. Metals can also influence the expression of genes and, therefore, the metabolism of cells in biofilms [14].

In prokaryotes, metals can induce biofilm formation or dispersion. Some bacteria respond to metal exposure by promoting surface adherence and increasing biofilm generation and persistence. In other organisms, the presence of metals can prevent changes from planktonic to sessile lifestyle. Therefore, biofilms need to be already formed when metal stress arises in order to be a resistance mechanism [14].

Changes in biofilms architecture have also been reported in the presence of metals. For example, *Archaeoglobus fulgidus* biofilms show different morphology in the presence of NaCl, where they form a cumulus of cells and fiber material, whereas high pH induces the formation of extensive layers [13].

The response to metals in archaeal biofilms has not been extensively reported, except in archaea such as *Sulfolobus metallicus* [15] and *Acidianus* spp. [16,17], both involved in biomining operations.

In archaea, few metal resistance mechanisms have been reported, consisting mostly of active detoxification systems, such as the Cop system, or PitA and Pho84 homologs [18,19,20]. PolyP is not only involved in biofilm formation [12], but also may be considered as another heavy metal resistance system [18,21].

Whether biofilm formation could represent a form of protection against Cu remains unsolved. This work aims to explore the effect of the presence of Cu in biofilm formation, and the apparent role this form of life has in Cu resistance of *S. solfataricus,* a model archaeon for which genetic tools are available.

## 2. Materials and Methods

### 2.1. Strains and Planktonic Growth Conditions

*S. solfataricus* M16 and derived strains were grown at 75 °C by shaking at 150 rpm in Brock medium: 1.3 g/L (NH_4_)_2_SO_4_, 0.28 g/L KH_2_PO_4_, 0.25 g/L MgCl_2_ × 7 H_2_O, 0.07 g/L CaCl_2_ × 2 H_2_O, 0.02 g/L FeCl_2_ × 4 H_2_O, 1.8 mg/L MnCl_2_ × 4 H_2_O, 4.5 mg/L Na_2_B_4_O_7_ × 10 H_2_O, 0.22 mg/L ZnSO_4_ × 7 H_2_O, 0.06 mg/L CuCl_2_ × 2 H_2_O, 0.03 mg/L Na_2_MoO_4_ × 2 H_2_O, 0.03 mg/L VoSO_4_ × 2 H_2_O, and 0.01 mg/L CoCl_2_ × 6 H_2_O, pH 3, and supplemented with 0.1% (*w*/*v*) N-Z amine (Sigma-Aldrich^®^, Merk KGaA, Burlington, MA, USA), 0.2% (*w*/*v*) glucose, and 0.01 mg/mL uracil only in the case of uracil autotrophs. For overexpression of PPX, 0.2% (*w*/*v*) D-arabinose (Sigma-Aldrich^®^, Merk KGaA, Burlington, MA, USA) was added.

### 2.2. Microtitration Plate Assays

*S. solfataricus* cells were grown in Brock medium, pH 3.5 and supplemented with 0.1% N-Z-amine and 0.2% D-arabinose or uracil, in 96-well microtitration plates covered with a gas permeable sealing membrane (Breathe-Easy, Diversified Biotech, Boston, MA, USA) at 75 °C inside a humidity chamber to avoid evaporation. The initial OD_600nm_ of cultures was 0.03. After 2 days, supernatant was removed and filtered. Filtered medium was placed back, and different Cu concentrations were added. Plates were sealed again with a membrane and placed back into the incubator. After 2 more days of growth, plates were cooled down and supernatant was placed in a new plate to determine OD_600nm_ with an Epoch luminometer (BioTek instruments, Agilent Technologies, Inc., Santa Clara, CA, USA). Ten µL of 0.5% crystal violet solution (CV) was added to each well containing biofilm, followed by incubation for 10 min at RT. Sessile cells were washed with Brock pH 5 medium, and CV attached to the biofilm was released by using 200 µL of 30% acetic acid. The released CV was measured at OD_570nm_. OD_570_/OD_600_ correlation index was used to determine biofilm formation efficiency.

### 2.3. Epifluorescence Microscopy and Confocal Laser Microscopy

For epifluorescence and confocal laser microscopy (CLM), cells were grown in 35 mm Petri dishes (µ-dishes; Ibidi; Martinsreid) as previously described [5]. Culture medium contained 0.5 mM CuSO_4_ and was exchanged every 24 h. After 3 days, supernatant was exchanged for 2 mL Brock pH 5 medium, and biofilm was stained with 3.6 µL of DAPI (500 µg/mL) and 15 µL ConA-fluorescein (5 mg/mL) for epifluorescence microscopy. Additionally, IB4-Alexa568 (lectin IB4 from *Griffonia simplicifolia*) (Invitrogen^TM^, Thermo Fisher Scientific, Waltham, MA, USA) was used for CLM. Staining was performed for 30 min in the dark at room temperature.

Biofilms were observed under microscopes from ZEISS: TIRF Observer 1 (Carl Zeiss Microscopy GmbH, Jena, Germany) with 100× objective for epifluorescence, and Observer 1 with 63× objective for CLM. Analysis of images was carried on in Fiji 2.12.0 (ImageJ) [22] and Carl Zeiss Zen (Blue Edition) software. Fluorescent signal quantification was performed with Fiji; the values represent the mean of the fluorescent signal in the histogram.

### 2.4. Adhesion Assays

For adhesion assays, 40 mL of *S. solfataricus* M16 cultures were grown in a 100 mL Schott flask with a glass slide inside. Initial OD_600nm_ was 0.03, and cultures were grown at 75 °C and 150 rpm. After 24 h, the glass slide was removed, washed twice with Brock medium pH 5, and attached cells were fixed with 4% formaldehyde dissolved in Brock medium pH 5. Cells and EPS were stained with 6 µL DAPI (300 µg/mL) and 15 µL (5 mg/mL) ConA dissolved in 1 mL Brock medium pH 5 for 30 min in the dark. Washing steps were performed twice with Brock medium, and slides were subsequently air-dried. Cells on the underside of the slides were removed with 70% ethanol. Finally, slides were observed using a TIRF Observer 1 ZEISS microscope with a 100× objective. Ten pictures from different fields were taken from three biological replicates, and processed with Fiji (ImageJ) [19].

### 2.5. Total RNA Extraction and cDNA Synthesis

For biofilm samples, cells were grown in large 150 mm Petri dishes starting at an initial OD_600nm_ of 0.03 in Brock medium with supplements. For PolyP (-) strain, D-arabinose was added. Cells were grown for 3 days at 75 °C with no agitation inside a humidity chamber, exchanging medium after the first 24 h. For controls, after 3 days, biofilm supernatants were removed and cells from biofilms were washed twice with Brock basic medium, scraped from the bottom with a cell scraper, and 20 mL of fresh medium was added. For biofilm exposed to Cu, after 2 days of biofilm formation as described above, media was filtrated, Cu was added and cell incubation continued for 24 h.

Planktonic cells were grown in the same conditions, but in Erlenmeyer flasks. CuSO_4_ was added to cultures at the mentioned concentrations for either 4 h or 24 h.

Biofilms and planktonic cells were harvested by centrifugation (7700× *g* for 15 min). Cell pellets, of 10 mg wet weight, were washed three times with Brock medium and lysed as previously described [20]. RNA was extracted using TRIzol (Invitrogen^®^) as described by the manufacturer. Remaining DNA was eliminated by adding 40 U of TURBO DNA-free DNase (Invitrogen™, Thermo Fisher Scientific, Waltham, MA, USA) following the manufacturer’s instructions. Quality of RNA was checked with agarose gel electrophoresis and PCR to confirm absence of DNA. For reverse transcription, 0.8 µg of total RNA was used for cDNA synthesis. Reaction was made by using ImProm-II ((Promega, Madison, WI, USA), 0.5 µg of random hexamers ((Promega, Madison, WI, USA), and 3 mM MgCl_2_ for 1 h at 42 °C. Two technical replicates of qPCR reactions were conducted for each three biological replicate of every experimental condition.

### 2.6. Primer Design and Real-Time RT-PCR

Primers for qRT-PCR were designed using Primer3 software version 4.1.0 and the annotated genome of *S. solfataricus P2*.

Gene expression was analyzed with a 96-well PikoReal-Time PCR System (Thermo Scientific, Waltham, MA, USA). Five µL of KAPA SYBR^®^ FAST 2X (Sigma-Aldrich^®^, Merk KGaA, Burlington, USA) was used, along with 0.2 µL of each primer, and 0.5 µL of a 1:20 dilution of the cDNA.

Efficiency of each pair of primers was calculated from the average slope of a linear regression curve, constructed from qPCRs using a 10-fold dilution series (10 pg–10 ng) of *S. solfataricus* M16 chromosomal DNA as template. Cq values (quantification cycle) after 40 cycles were automatically determined by PikoReal software 2.1 (Thermo Scientific). Cq values of each transcript of interest was standardized to Cq value of housekeeping genes *16s* rRNA and *Rps2P*. At least 3 biological replicates of each condition, and 2 technical replicates per qPCR reaction, were performed.

## 3. Results

### 3.1. Copper Induces Biofilm Formation in S. solfataricus M16

To test whether Cu influenced biofilm formation or dispersion, *S. solfataricus* biofilms were grown for 2 days, after which planktonic cells were eliminated from the media by filtration. Subsequently, the biofilm was exposed to different Cu concentrations below the reported MIC of 2 mM [21]. After 48 h, the OD_600_ of supernatant was measured and the crystal violet assay was performed to determine biofilm biomass (OD_570_). Biofilm formation was expressed as the OD_570_/OD_600_ index.

As seen in Figure 1A, there were differences in OD_600_ depending on Cu concentrations. At 1 mM Cu, the OD_600_ was significantly higher compared with control without Cu.

Regarding biofilm formation, biomass increased at low Cu concentrations (0.25, 0.5, and 0.75 mM Cu) compared with control, reaching a maximum when exposed to 0.5 mM Cu (Figure 1B). Next, a concentration of 1 mM Cu caused the opposite effect, leading to a reduction in biofilm biomass (Figure 1B), but as already mentioned, an increase in planktonic cells (Figure 1A).

Therefore, Cu induced biofilm formation at low concentrations, with a maximum at 0.5 mM Cu, after which, biofilm amounts decreased (Figure 1C).

### 3.2. Cu Affects Biofilm Morphology

Confocal laser microscopy (CLM) was performed to observe differences in biofilm appearance when exposed to Cu. Biofilms were grown on µ-dishes with Brock medium supplemented with N-Z amine, glucose, and uracil, with or without 0.5 mM Cu. The medium was exchanged every 24 h. As previously described, cell DNA was stained with DAPI. ConA and IB4 lectins were used to stain α-manopiranosyl and α-glucopiranosyl, and α-D-galactosyl residues, respectively [12]. One representative image from each condition can be seen in Figure 2. Additional images are shown in the Appendix A, Appendix A.

As seen in the DAPI channel (Figure 2A), the biofilm generated in the presence of Cu appeared to have more cells accumulated compared with control conditions. This was corroborated by quantification of the DAPI signal, as seen in the microscopy images in Figure 3. These cells seem to aggregate, forming clusters of cells along with a biofilm matrix (Figure 2A, merge channel).

As per the *z*-axis of images, biofilms formed in presence of Cu are thinner (Figure 2B). In the absence of Cu, biofilms surpass 10 μm, while when exposed to 0.5 mM Cu, biofilms barely reached 10 μm. Nevertheless, there seems to be more cells, and these appear to be packaged in a more compact distribution in the biofilm in the presence of metal (Figure 2B).

The identity and quantity of exopolysaccharides was also different, as can be seen in the lectins channel (ConA and IB4). In the presence of Cu, a decrease in the green signal (ConA, α-manopiranosyl and α-glucopiranosyl residues) can be seen, with an increase in the red signal (IB4, α-D-galactosyl residues) (Figure 2A, ConA and IB4). The increase in the IB4 signal was significant in the condition with Cu, compared with the control (Figure 3).

Altogether, Cu affects biofilm morphology, leading to a more compact biofilm with different exopolysaccharide patterns than *S. solfataricus* biofilm in the absence of Cu.

### 3.3. Cu Influences Archaellum Expression and Cell Adherence

Surface adherence is one of the initial steps for biofilm formation, and in *S. solfataricus,* it depends on the archaella [7,23]. To test whether Cu could induce cell adherence, cells were grown in shaking cultures with different Cu concentrations, with a glass slide inside. After 24 h, the number of cells attached to the slide was analyzed under the microscope using DAPI and ConA stains.

In Figure 4A, the mean number of cells per field can be observed at different Cu concentrations. Even though the difference between bars is not statistically significant, on average, there were more cells at 0.25 mM Cu than any other concentration. Interestingly, at 0.5 mM there were fewer cells than in the control condition.

Representative pictures with 0.25 and 0.5 mM Cu are shown in Figure 4B. As is evident, cells tend to form clusters in the presence of Cu, as observed in biofilm CLM images. In Appendix A, the control condition and other Cu concentrations can be seen.

Transcriptional levels of *arlB* were also determined in planktonic cells exposed to different Cu concentrations and two time points: 1 h and 24 h. As seen in Figure 4C, Cu increasingly induced *arlB* expression proportionally with the concentration of metal. This effect was even more apparent after 24 h of exposure.

Cu, therefore, induced transcription of an archaellum component in planktonic cells and altered cell attachment to the surface.

### 3.4. Transcriptional Levels of Genes Related to Biofilm in the Presence of Cu

The expression levels of two other genes related to biofilm formation were measured by using qPCR. Figure 5 shows the transcript levels expressed as log_2_ (fold change) for *SSO1101* and *SSO3006*. Gene *SSO1101* is homologous to *saci_1223*, a general regulator that is thought to positively influence biofilm formation [9], while *SSO3006* codes for an α-mannosidase involved in exopolysaccharide synthesis and cell attachment during biofilm formation [24].

As seen in Figure 5A, the presence of Cu caused repression of *SSO1101* in planktonic cells after 1 h and 24 h exposure, at all tested Cu concentrations. In biofilms, the expression of *SSO1101* did not show significant changes at 0.25 and 0.75 mM Cu. Interestingly, gene expression was downregulated at 0.5 and 1 mM Cu in biofilm cells.

On the other hand, α-mannosidase *SSO3006* (Figure 5B) showed different expression levels at the tested conditions. After 24 h of exposure to Cu, a slight increase proportional to the metal concentration was seen in planktonic cells. In sessile cells (biofilm), levels of α-mannosidase remained almost unchanged, except at 0.75 mM. It has been previously seen that overexpression of *SSO3006* caused a decrease in exopolysaccharides in *S. solfataricus* biofilm [24].

### 3.5. Transcriptional Levels of CopA Are Higher in Planktonic Than in Biofilm Lifestyle

As mentioned before, CopA is the major Cu resistance determinant in *S. solfataricus,* and its expression is regulated by intracellular Cu concentration [25]. Therefore, *copA* levels were measured to determine whether this induction of expression occurs in biofilms, as it happens in planktonic cells.

Interestingly, *copA* levels were higher in planktonic cells when compared to sessile cells exposed to the same Cu concentration (Figure 6). While in planktonic cells *copA* levels increased with Cu concentration, *copA* expression in sessile cells at 0.5 mM Cu was less than that seen at 0.25 or 0.75 mM Cu. This result supports the idea that biofilms might confer protection against Cu to this archaeon, particularly, given that *copA* expression was decreased while biofilm formation was at its maximum at 0.5 mM Cu (Figure 1).

### 3.6. Cu Did Not Induce Biofilm Formation in a PolyP (-) Strain

It was previously shown that PolyP is involved in biofilm formation [12]. Therefore, the effect of Cu on biofilm morphology in a PolyP (-) mutant [21] was investigated. As seen by CLM in Figure 7, the same Cu concentration already used (0.5 mM) did not induce the previously described effects seen on biofilms in the wild-type strain. In the PolyP (-) strain, sugar residues detected by lectins (ConA and IB4, Figure 7A) were lower, and biofilm height was even shorter in the presence of Cu (Figure 7B, Appendix A and Appendix A).

Analyzing the fluorescent signal of the images, it can be seen that there are no significant changes in the levels of any of the signals (Figure 8).

It has been previously reported that PolyP (-) mutants generate less biofilm than background counterpart strains [12].

Altogether, in PolyP-deficient mutants, Cu could not induce biofilm formation, as it occurs in *S. solfataricus* M16, at the tested concentrations.

## 4. Discussion

It is well known that biofilm forms of life can offer advantages against several stress factors, one of which is metals [26]. Sessile cells from biofilms are exposed to variable conditions of pH, oxygen, and nutrients inside biofilm structures, whereas this does not occur in planktonic cells [26,27]. This leads to heterogeneity amongst the cell population, and it is believed to enhance stress resistance. Several layers of biofilm also cause cells in lower layers to be less exposed to these factors. The EPS forming the matrix generally possesses a negative charge, and therefore can trap metal cations, preventing their entrance into cells [14]. Archaea can also modify their S-layer in response to stress by changing glycosylation patterns of their proteins [28,29].

It has been described that bacteria, such as *Halobacterium salinarum,* are more resistant to Cu and Ni in biofilm than in planktonic lifestyle [30]. Archaeal biofilm formation on mineral surfaces has been studied before [15,16,17,31,32,33], although responses to Cu or other metals of cells in this form of life have not been described.

In the experiments reported here, the presence of certain Cu concentrations increased the amounts of biofilm formed by *S. solfataricus*, reaching a maximum at 0.5 mM Cu.

Biofilm morphology was studied after 3 days in the presence of 0.5 mM Cu using CLM. In these biofilms, the structure formed by cells in the presence of metal was denser, while changes in polysaccharides were also seen. By using microscopy, considerably lower amounts of α-manopiranosyl and α-glucopiranosyl residues (labeled with ConA) were seen in the presence of Cu compared with control conditions (Figure 2 and Figure 3). On the other hand, an increase in α-D-galactosyl residues (IB4 signal) was also detected. Minding these changes, in general, smaller amounts of sugars were observed in the presence of Cu. It is known that EPS can trap metals on their negative charges, and these changes in *S. solfataricus* biofilm seem to indicate an adaptation of extracellular glycosylation motifs to metal concentration in order to protect the cells. In line with this, the expression of *cop*A, which is responsive to intracellular Cu concentrations [34], was also decreased at 0.5 mM Cu in biofilm cells compared with planktonic cells (Figure 6). These observations support the idea that biofilms could offer a form of protection to cells against metal stress, by modification of the glycosylation patterns or the released EPS.

In the response of *S. acidocaldarius* to butanol, another stressor agent, polysaccharide amounts increased in biofilms [35]. Opposed to what we observed, a predominance of α-manopiranosyl and α-glucopiranosyl residues, versus α-D-galactosyl residues, was seen [35]. In *S. acidocaldarius,* it is known that mutants for the Agl3 enzyme, which is also related to protein glycosylation, have trouble growing in the presence of NaCl [36]. This also takes place in Agl6 mutants [37]. *H. volcanii* also showed changes in its S-layer protein glycosylation patterns as a response to environmental stress [29]. These changes suggest that glycosylation of proteins on the cell surface, as well as changes in exopolysaccharides released to the biofilm matrix, would allow archaeons to adapt to stress challenges. It has also been suggested that EPS from archaeal biofilms could facilitate the transport of metabolites between cells [17,38]. Further studies isolating proteins from the S-layer or characterizing EPS could be conducted to obtain a better understanding of these changes.

At 1 mM Cu, biofilms seemed to disperse, which might indicate that the microorganism prefers other forms of life to cope with this metal concentration. In previous MIC assays in liquid culture, 1 mM Cu caused a 25% reduction of growth in planktonic cells compared with control conditions without metal [21]. The rise in the number of planktonic cells at this Cu concentration could possibly result from cells escaping the biofilm at high metal concentrations. In agreement with this idea, a decrease in OD_570_ is seen at the same Cu concentration (Figure 1B).

In bacteria, it has been described that metals can also disrupt biofilms. That is the case of *Acidithiobacillus ferrooxidans* ATCC23270 in response to Cu when growing on elemental sulfur [39], and *Burkholderia multivorans* in response to nickel and calcium [40]. In the last examples, this mechanism occurs by the inhibition of acyl-homoserine lactone quorum sensing, a system that seems to be absent in Sulfolobales.

As stated before, biofilm formation in archaea remains poorly studied. In *S. acidocaldarius*, the presence of 1.5% (*v*/*v*) 1-butanol caused biofilm formation and changes in biofilm architecture [35], whereas 1% (*v*/*v*) 1-butanol was the maximum concentration at which planktonic growth was seen.

Several genes are involved in biofilm formation. In this study, *SS1101* and *SSO3006* transcriptional levels after Cu exposure were studied in planktonic and sessile cells. *SSO3006* codes for an α-mannosidase involved in biofilm formation. In previous studies, deletion of this gene led to an increased amount of exopolysaccharides, while its complementation via plasmid led to a decrease in the quantity of these sugars, as well as changes in the composition of *S. solfataricus* biofilm [24]. In the present report, while at 0.5 mM Cu, expression levels of this gene remained unchanged, it was interesting to observe a significant increase of *SSO3006* in the biofilms at 0.75 mM Cu. Although we did not explore the morphology of biofilms at concentrations different than 0.5 mM, it would be of interest to further analyze possible changes. Furthermore, increased expression levels of the *SSO3006* gene could explain, in part, the decreased biofilm generation at 0.75 mM Cu (Figure 1) when compared to that seen at 0.5 mM Cu. It is worth noting that there are other genes related to the synthesis of exopolysaccharides that were not analyzed here.

Interestingly, *SSO1101* was downregulated in the presence of 0.5 mM Cu, where an increase in biofilm formation took place. In biofilms exposed to 1-butanol, this gene was also downregulated [35]. *SSO1101* is homologous to *saci_1223*, a general regulator belonging to the Lrs-14 family, whose deletion impairs biofilm formation [9]. This gene was upregulated in biofilms, compared to planktonic cells, in transcriptomic and proteomic studies in *S. solfataricus*, as well as its homologs in *S. acidocaldarius* and *S. tokodaii* [6]. These observations might indicate a general downregulation response of this transcriptional regulator to confront stress. Target genes regulated by *SSO1101* and homologs remain unknown, whereas it has been proposed that one of its targets could be the *vapBC14* operon, which was also described to be upregulated in *S. acidocaldarius* biofilms [11]. The proteins VapB14 and VapC14 coded in this operon influence biofilm formation as well [11].

The exposure of planktonic cells to Cu caused an increase in transcriptomic levels of *arlB* (Figure 3). This gene codes for the filament component of the archaellum, being responsible for motility in archaea [41]. Although archaella and flagella are different structures, the former rather resembling type IV pili, it is interesting to notice that upregulation of flagellar assembly machinery was seen in response to Cu in the moderate thermophilic bacteria *Acidithiobacillus caldus* [42]. In *S. solfataricus*, while the archaellum is still involved in cell motility, it also helps with initial attachment during biofilm formation [7,23]. In experiments testing the adhesion of cells to glass when exposed to Cu (Figure 3), no conclusive results were seen. It is difficult to determine whether overexpression of this appendage component induces motility or attachment in response to Cu, and furthermore, how this switch in function is accomplished. Additionally, the state of glycosylation of archaella and S-layer proteins could also influence this phenomenon.

Regarding the Cop system, the main response mechanism to Cu in *S. solfataricus*, an increase in *copA* transcripts, was observed by qPCR in planktonic cells. As expected, this upregulation correlated with an increase in external Cu concentration (Figure 6). This induction of expression was higher for planktonic cells than sessile cells at the same Cu concentration (Figure 6). It was interesting to notice that upregulation in the biofilm was the lowest in the presence of 0.5 mM Cu, where Cu was also inducing biofilm formation. These results suggest that in the sessile lifestyle, less Cu might enter cells thanks to protection by external sugars and modification of the S-layer, as already mentioned.

Finally, Cu could not induce biofilm formation in a PolyP (-) mutant (Figure 7A). This could be related to the Cu concentration used in this experiment, which is close to the MIC value of 0.75 mM for this strain [21]. PolyP has been shown to play a role in biofilm formation [12], as well as in metal resistance [18,21]. Additional studies will be required to elucidate whether this effect is a product of one of these factors.

Altogether, evidence is provided here that Cu can promote biofilm formation in *S. solfataricus*. This form of life can protect cells from a toxic metal.

## 5. Conclusions

Several factors contribute to metal resistance, besides active detoxification systems. One of these is biofilm formation. Here, we explored the response of *S. solfataricus* to Cu in the biofilm form of life.

Biofilm formation assays and CLM images revealed changes in the formation, architecture, and exopolysaccharide patterns of biofilms in response to Cu. An increase in biofilm formation was seen, reaching a maximum at 0.5 mM Cu. At this metal concentration, the biofilm appeared to be thinner, and cells were packaged in a more compact distribution than the control. Sugars residue patterns also changed, with a decrease in the amounts of α-manopiranosyl and α-glucopiranosyl, while D-galactosyl residues increased. This suggests modification of external glycosylation patterns, probably in the S-layer or surface appendages, or even EPS composition. This was accompanied by a decrease in the expression of *copA*, indicating less exposure of sessile cells to Cu.

It will be of interest to analyze possible changes in the morphology of biofilms at different concentrations of Cu and other metals, as well as the EPS composition.

In summary, it was seen that biofilm formation is involved in metal response in this archaeon, potentially furnishing defense against metal-induced stress. According to what was observed in this study, the mentioned biofilm changes would enable *S. solfataricus* to adapt to stressful challenges, offering protection to cells.

The results presented here provide new insights into the response of *S. solfataricus* to a toxic metal. Mutants in *S. solfataricus* or other Sulfolobales with genetic tools available could help to determine the exact mode of regulation of archaeal biofilm formation to resist Cu and other types of stress. A better understanding of these phenomena could be of importance in order to obtain strains with improved resilience for industrial use. Furthermore, whether these findings can be extrapolated to other thermoacidophiles, especially those utilized in biomining processes, remains to be analyzed.

## Figures and Tables

**Figure 1 microorganisms-11-01421-f001:**
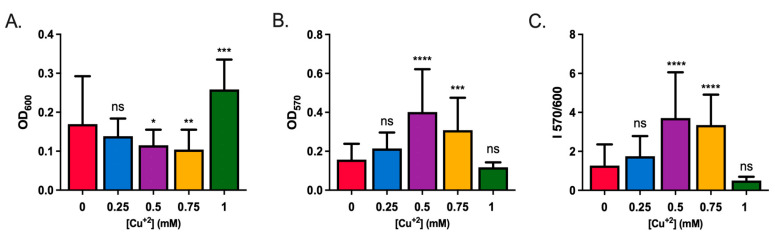
Copper induces biofilm formation in *S. solfataricus*. Biofilms were grown for 2 days in 96-well plates before exposure to Cu for 48 h. (**A**) Growth of planktonic cells (OD_600_). (**B**) Biofilm mass stained with crystal violet (OD_570_). (**C**) OD_570_/OD_600_ index. ANOVA test: **** indicating *p* ≤ 0.0001, *** *p* ≤ 0.001, ** *p* ≤ 0.01, * *p* ≤ 0.05, and ns: not significant.

**Figure 2 microorganisms-11-01421-f002:**
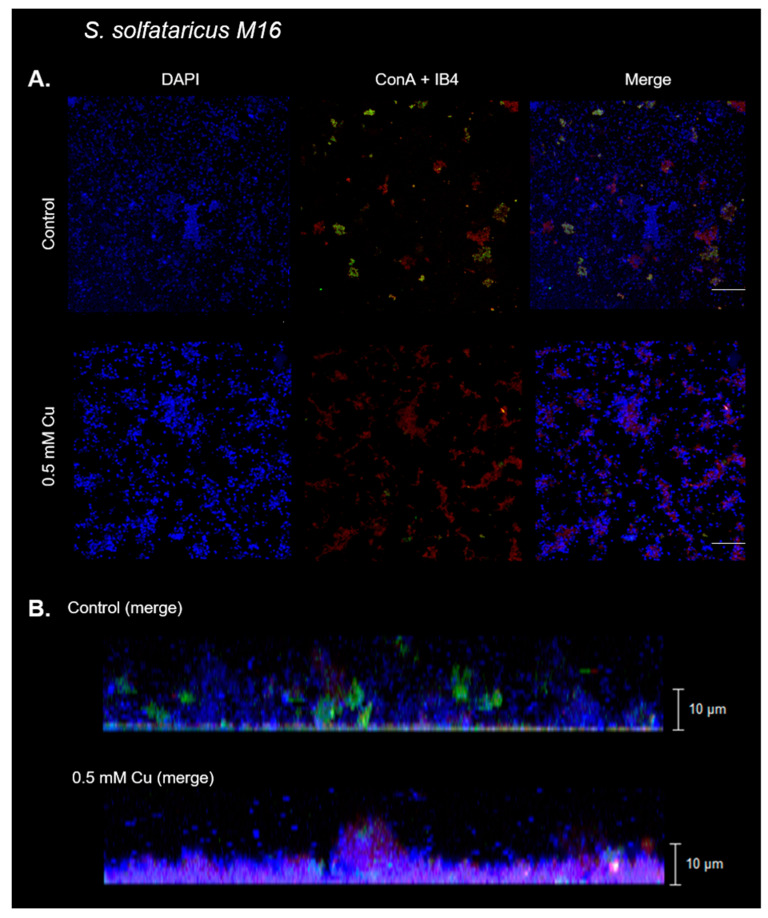
Copper affects biofilm morphology in *S. solfataricus*. Representative confocal laser microscopy images of *S. solfataricus* biofilms exposed to 0.5 mM Cu, and control (nonexposed) biofilm. DNA from cells were stained with DAPI (blue signal). Lectins IB4 (red signal) and ConA (green signal) were used to stain α-D-galactosyl, and α-manopiranosyl and α-glucopiranosyl residues, respectively. (**A**) XY view from biofilm and merge of three channels. White bar corresponds to 20 µm. (**B**) Z stack with the merge of three channels.

**Figure 3 microorganisms-11-01421-f003:**
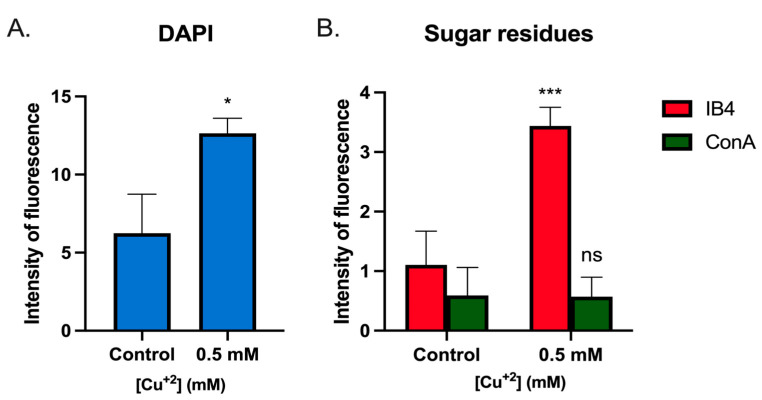
Analysis of fluorescence signals in *S. solfataricus* biofilms. Fluorescence signals of the obtained CLM images of biofilms formed in the presence or absence of Cu were extracted from each individual channel and analyzed. Values represent the mean of the signal in the sample. (**A**) DAPI. (**B**) Sugar residues detected with lectins tagged with IB4 and ConA. Statistical analysis with one-way ANOVA: *** *p* ≤ 0.001, * *p* ≤ 0.05, and ns: not significant.

**Figure 4 microorganisms-11-01421-f004:**
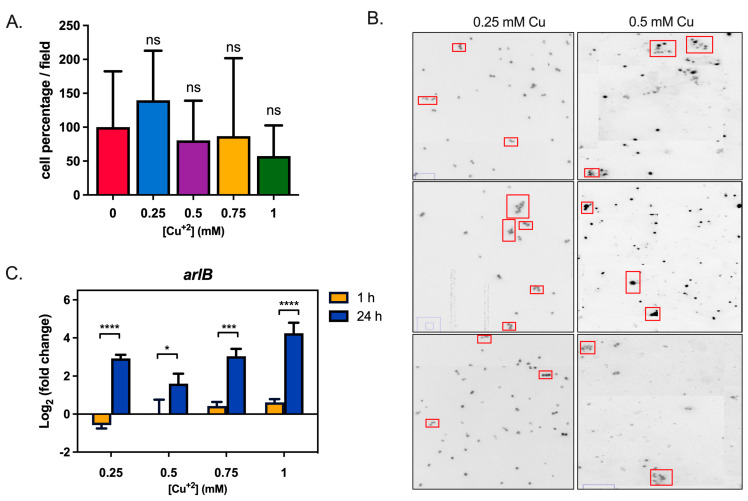
Copper does not increase adhesion to the glass surface, but affects archaellum expression in *S. solfataricus*. (**A**) Mean of cells per field in adhesion experiments using the indicated Cu concentrations. ANOVA test: ns: not significant. (**B**) Representative images of cell clusters from adhesion experiments, indicated by red boxes. (**C**) Analysis of *arlB* transcripts levels via qPCR. Statistical analysis with one-way ANOVA with Holm–Sidak correction: **** indicating *p* ≤ 0.0001, *** *p* ≤ 0.001, * *p* ≤ 0.05, and ns: not significant.

**Figure 5 microorganisms-11-01421-f005:**
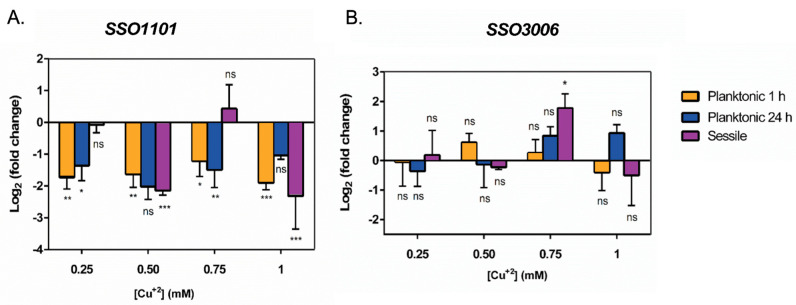
Levels of genes related to biofilm formation in *S. solfataricus*. Levels of gene transcripts for (**A**) SSO1101 and (**B**) SSO3006 in sessile (biofilm) and planktonic cells exposed to the mentioned Cu concentrations for 1 h or 24 h. Transcript levels were normalized against control conditions without Cu. Statistical analysis with one-way ANOVA with Holm–Sidak correction: *** *p* ≤ 0.001, ** *p* ≤ 0.01, * *p* ≤ 0.05, and ns: not significant.

**Figure 6 microorganisms-11-01421-f006:**
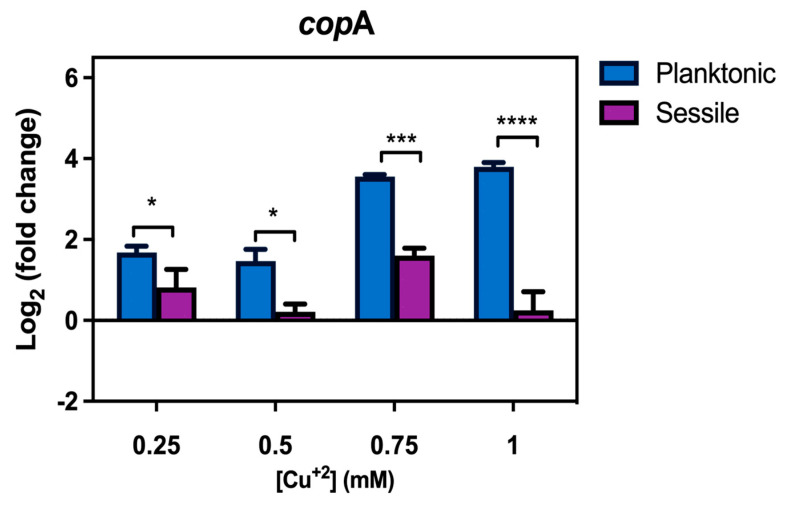
*CopA* is less transcribed in sessile cells when compared to planktonic cells at the same Cu concentration. Transcript levels of sessile (biofilm) and planktonic cells exposed to the mentioned Cu concentrations for 24 h. Transcript levels were normalized against control conditions without Cu. Statistical analysis was one-way ANOVA with Holm–Sidak correction: **** indicating *p* ≤ 0.0001, *** *p* ≤ 0.001, * *p* ≤ 0.05, and ns: not significant.

**Figure 7 microorganisms-11-01421-f007:**
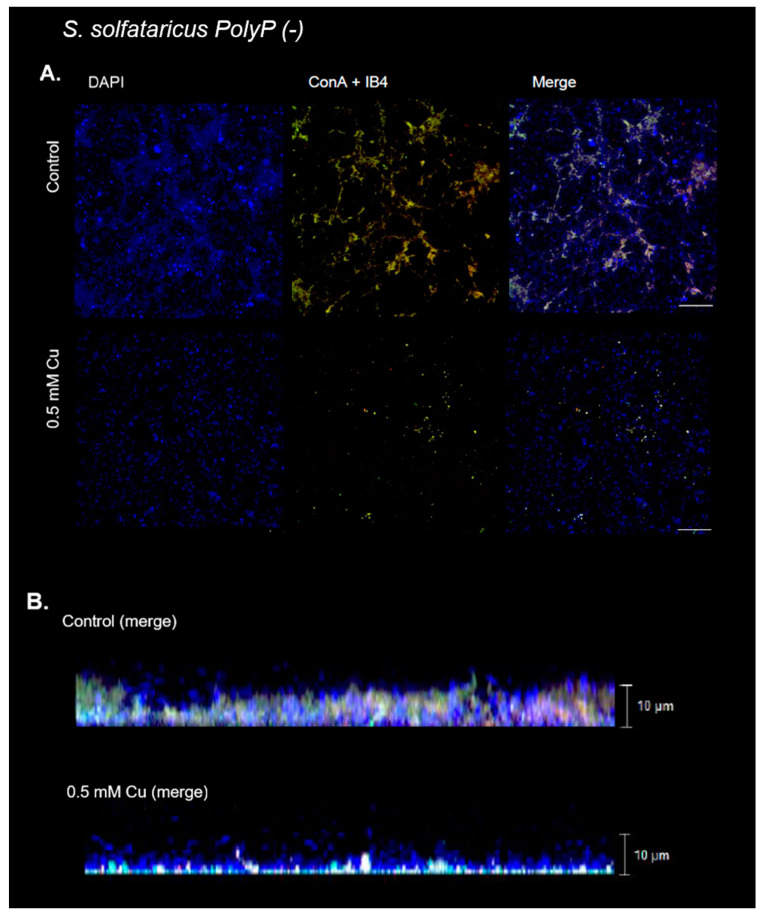
Cu does not promote biofilm formation in PolyP (-) strain at 0.5 mM Cu. Representative confocal laser microscopy images of *S. solfataricus* PolyP (-) biofilms exposed to 0.5 mM Cu and control (nonexposed) biofilm. DNA from cells was stained with DAPI (blue signal). Lectins IB4 (red signal) and ConA (green signal) were used to stain α-D-galactosyl, and α-manopiranosyl and α-glucopiranosyl residues, respectively. (**A**) XY view of biofilm and merge of three channels. White bar corresponds to 20 µm. (**B**) Z stack with the merge of three channels.

**Figure 8 microorganisms-11-01421-f008:**
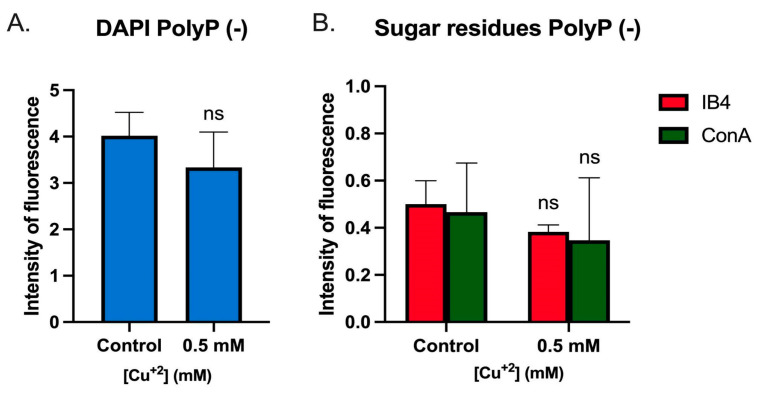
Analysis of fluorescence signals of *S. solfataricus* PolyP (-) strain biofilms. Fluorescence signals of the obtained CLM images of biofilms formed in the presence or absence of Cu were extracted from each individual channel and analyzed. Values represent the mean of the signal in the sample. (**A**) DAPI. (**B**) Sugar residues detected with lectins tagged with IB4 and ConA. Statistical analysis with one-way ANOVA: ns: not significant.

## Data Availability

Data available upon request to the corresponding author.

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
