# Peer review of "Sessile Lifestyle Offers Protection against Copper Stress in Saccharolobus solfataricus"

_microorganisms, 2023, doi:10.3390/microorganisms11061421_

Round 1

Reviewer 1 Report

1.The individual bars in Figure 1 are less distinguishable and a different colour is recommended.

2.Fig. 3a and c without top and right borders.

3.Why are the patterns in Fig. 2a and Fig. 6a opposite but the patterns in Fig. 2b and Fig. 6b the same?

4.Figures 4 and 5 do not have a top and right border and the colours are not sufficiently differentiated.

5.The conclusions are too few and it is recommended that more be written.

No problems overall, a few more touches are recommended.

Reviewer 2 Report

Abstract:

Kindly mention why copper stress is an important topic to study in archaea. The methods part could be condensed and presented more clearly. The findings section could also be presented more clearly.

Introduction

The introduction does not clearly state a problem statement, it does not present a specific research question or problem that the paper aims to address.

Please shorten the subtitles in the results section.
In conclusion, please summarize the main findings, and link the findings to the research question. The conclusion should also discuss the limitations of the study and suggest directions for future research. 

The language in the manuscript could be simplified to improve its clarity. Some sentences are quite complex and may be difficult to understand. Consider revising the text to make it more accessible and easier to follow.

Reviewer 3 Report

In this study, authors evaluated the effect of copper concentration on planktonic and sessile cells of Saccharolobus solfataricus. The research work shows interesting results and demonstrates that the biofilm matrix protects cells from environmental stressors, contributing to increase the resistance of microorganisms to metals. 

The methodological approach used in the study included CLM data; however, post-image analysis was not conducted. Therefore, it is recommended to extract fluorescence data and perform statistical analysis to evaluate if differences in biofilm structure in the presence of copper were significant.   
